# Can Post-Activation Performance Enhancement (PAPE) Improve Resistance Training Volume during the Bench Press Exercise?

**DOI:** 10.3390/ijerph17072554

**Published:** 2020-04-08

**Authors:** Michal Krzysztofik, Michal Wilk, Aleksandra Filip, Piotr Zmijewski, Adam Zajac, James J. Tufano

**Affiliations:** 1Institute of Sport Sciences, Jerzy Kukuczka Academy of Physical Education in Katowice, 40-065 Katowice, Poland; m.krzysztofik@awf.katowice.pl (M.K.); a.filip@awf.katowice.pl (A.F.); a.zajac@awf.katowice.pl (A.Z.); 2Jozef Pilsudski University of Physical Education in Warsaw, 00-809 Warsaw, Poland; piotr.zmijewski@insp.waw.pl; 3Faculty of Physical Education and Sport, Charles University, 16000 Prague, Czech Republic; tufano@ftvs.cuni.cz

**Keywords:** strength-endurance, repetition, time under tension, power output, bar velocity

## Abstract

*Background*: The aim of the present study was to evaluate the effects of post-activation performance enhancement (PAPE) on resistance training volume during the bench press exercise (BP). The study included 12 healthy strength-trained males (age 25.2 ± 2.1 years, body mass 92.1 ± 8.7 kg, BP one-repetition maximum (1RM) 28.8 ± 10.5 kg, training experience 6.3 ± 2.1 years). *Methods*: The experiment was performed following a randomized crossover design, where each participant performed two different exercise protocols with a conditioning activity (CA) consisting of the BP with three sets of three repetitions at 85% 1RM (PAPE), and a control without the CA (CONT). To assess the differences between PAPE and CONT, the participants performed three sets of the BP to volitional failure at 60% 1RM. The differences in the number of performed repetitions (REP), time under tension (TUT), peak power output (PP), mean of peak power output (PP_MEAN_), mean power output (MP), peak bar velocity (PV), mean of peak bar velocity (PV_MEAN_), and mean bar velocity (MV) between the CONT and PAPE conditions were examined using repeated measures ANOVA. *Results*: The post-hoc analysis for the main condition effect indicated significant increases in TUT (*p* < 0.01) for the BP following PAPE, compared to the CONT condition. Furthermore, there was a significant increase in TUT (*p* < 0.01) in the third set for PAPE compared to the CONT condition. No statistically significant main effect was revealed for REP, PP, PV, PP_MEAN_, PV_MEAN_, MP, and MV. *Conclusion*: The main finding of the study was that the PAPE protocol increased training volume based on TUT, without changes in the number of preformed REP.

## 1. Introduction

Athletic performance requires a high level of various training components that can be developed through a sports conditioning program. Recently, much attention has been given to acute increases in exercise performance through methods that induce a post-activation performance enhancement (PAPE) evoked by prior muscle activity [1]. PAPE has been defined as a physiological phenomenon which acutely improves voluntary muscular performance (e.g., jumps and throws) following a conditioning activity (CA) which includes a single heavy loaded resistance exercise [2,3,4] or consecutive sets of a single resistance exercise [5,6]. In training practice, PAPE can be achieved by the use of a CA with a wide range of external loads, ranging from 70% of one-repetition maximum (1RM) [6] to even supramaximal loads of 110–130% 1RM [7]. Furthermore, training experience [8], individual strength level [3], as well as muscle groups involved in the activity may affect the magnitude of subsequent performance enhancement. 

To date, studies have focused on the acute changes in power output and velocity of movement during explosive activities (e.g., jumps and throws) directly after various types of CA [4,7]. However, little attention has been given to the potential impact of PAPE on training volume. To achieve a desired training volume, performing a certain number of repetitions (REP) per set, per exercise, and per session significantly affects the adaptive changes to resistance training [9,10]. However, the duration of a single REP is not always the same (it depends on the tempo of movement used), which is why, in addition to the number of performed REP in a set or a whole session, the time under tension (TUT) is considered as a variable describing training volume [11,12]. Furthermore, according to Wilk et al. [12,13], TUT during resistance exercise is a more accurate and credible indicator of the training volume performed compared to the number of performed REP. To date, only two studies have examined the effect of PAPE on resistance training volume of the upper body, yet only one of them considered TUT. Sevilmiş and Atalağ [14] indicated a significantly increased number of performed REP and TUT during a single set of the bench press exercise (BP) performed to volitional failure at 65% 1RM after a CA performed with eccentric only contractions at 120% 1RM, compared to control conditions. Furthermore, Alves et al. [15] also reported improved training volume evaluated by the total lifted load (REP x load) and the maximum number of performed REP after a CA.

However, it should be noted that most studies considering the acute effects of PAPE were assessed on the basis of only a single set of a post-activation exercise [7,16,17], while most resistance training guidelines recommend multiple sets of each exercise to achieve the desired adaptation [10]. To the best of our knowledge, only Alves et al. [15] evaluated the PAPE effect on resistance training volume during an upper-body exercise consisting of several sets. Alves et al. [15] showed that the PAPE effect increases total lifted load and the number of performed REP during three sets of the BP to volitional failure at 75% 1RM after a CA of three REP at 90% 1RM. However, there are no more studies confirming this effect, especially when using other training variables (external load, time rest, number of sets, tempo of movement), both during the CA and in the post-activation exercise. 

Furthermore, the effectiveness of PAPE during resistance exercise preformed to volitional failure is not only related to the volume of work, but also to the ability to maintain a high level of movement velocity and power output. Decreases in bar velocity during particular sets of an exercise have been accepted as a valid indicator of neuromuscular fatigue [18]. However, currently there is no research available on the impact of PAPE on exercise volume, with a simultaneous analysis of power output and bar velocity changes. Another factor in the optimal utilization of PAPE includes the determination of individual intra-complex rest intervals (IRT) between the CA and the post-activation exercise. According to Golas et al. [4], the IRT should be customized individually for each study participant. Recovery duration shows a large inter-individual variability that is associated with numerous factors such as strength level, training experience, and myotypology [4,19,20,21]. Therefore, the estimation of the individual IRT for each participant should be optimized, most likely using the trial and error method, experimenting with a rest interval from 2 to 10–12 min.

Since, numerous studies have confirmed the PAPE effect during intensive, short-duration activities, it would be interesting to investigate whether this phenomenon affects training volume during resistance exercise carried on to volitional failure. Furthermore, it would be interesting to investigate how PAPE protocols may attenuate the decrease in movement velocity and power output commonly observed during sets performed to volitional failure. Thus, the aim of the present study was to evaluate the effects of PAPE on training volume assessed by the number of performed REP and TUT during the BP exercise among strength trained participants. An additional aim of the study was to assess the impact of PAPE on power output and bar velocity during the BP exercise performed to concentric volitional failure. 

## 2. Materials and Methods 

The experiment was performed following a randomized crossover design, where each participant performed the following test protocols: one with a CA consisting of 3 sets of 3 repetitions at 85% 1RM (PAPE) and a control without the CA (CONT). Before the main experiment each participant performed two familiarization sessions, one with the 1RM test and the second with an individual intra-complex rest interval (IRT). The entire research procedure lasted 4 weeks with a one-week interval between each trial. During the experimental sessions, the participants performed 3 sets of the BP exercise to establish the maximum number of preformed REP to volitional failure with a load of 60% 1RM. The repetitions were performed with maximal velocity in the eccentric and concentric phases of movement. The following variables were registered: number of performed repetitions (REP), time under tension (TUT), peak power output (PP), mean of peak power output (PP_MEAN_), mean power output (MP), peak bar velocity (PV), mean of peak bar velocity (PV_MEAN_), and mean bar velocity (MV). All testing was performed in the Strength and Power Laboratory at the Academy of Physical Education in Katowice.

### 2.1. Participants

Twelve healthy strength-trained men participated in this study (age = 25.2 ± 2.1 years, body mass = 92.1 ± 8.7 kg, BP 1RM = 128.8 ± 10.5 kg), with a minimum 3 years of resistance training experience (6.3 ± 2.1 years). The inclusion criteria was a BP personal record of at least 120% of body mass. Participants were allowed to withdraw from the experiment at any moment and were free from any musculoskeletal disorders. The participants were instructed to maintain their normal dietary habits over the course of the study and not to use any supplements or stimulants for the duration of the experiment. They were informed about the benefits and potential risks of the study before providing their written informed consent for participation. The study protocol was approved by the Bioethics Committee for Scientific Research at the Academy of Physical Education in Katowice, Poland (10/2018), and performed according to the ethical standards of the Declaration of Helsinki, 1983.

### 2.2. Procedures

#### Familiarization Session

Two weeks before the main experiment, the participants performed 2 familiarization sessions, once per week. The first familiarization session included the 1RM test. During the second familiarization session, one week before the main experiment, IRT testing was performed [4]. The participants arrived at the laboratory at the same time of day as the upcoming experimental sessions and performed a standardized general and specific warm-up before each of the familiarization and main sessions. The warm-up protocol included 5 min cycling on a stationary ergometer (heart rate of around 130 bpm), followed by a general upper-body warm-up of 10 trunk rotations and trunk side-bends on each side, 10 internal and external rotary movements of the shoulders, and 10 push-ups. Next, the participants performed 15, 10, and 5 BP repetitions using 20%, 40%, and 60% of their estimated 1RM. The first test load was set to an estimated 80% 1RM and was increased by 2.5 to 10 kg for each subsequent attempt. This process was repeated until failure. During the 1RM test, 5 min rest intervals were given between each attempt and the 1RM was attained within 5 attempts. According to Wilk et al. [22], all trials during the 1RM test were performed with a constant duration in the eccentric phase (2 s). Hand placement on the barbell was set at 150% of the individual bi-acromial distance. The positioning of the hands was recorded to ensure consistent hand placement during all experimental sessions.

During the second familiarization session the IRT test was performed. The participants performed 2 repetitions of an explosive BP (60% 1RM) at baseline. After a 5 min rest interval they performed a CA consisting of 3 sets of 3 repetitions at 85% 1RM with 4 min rest intervals between each set. After 4, 8, 12 and 16 min of recovery, the participant performed a test similar to baseline (2 repetitions at 60% 1RM) to establish an optimal rest interval for each participant. An optimal individual IRT was selected as the best single repetition in peak power output in comparison to baseline [4]. 

### 2.3. Experimental Sessions

The general and specific warm-up for the experimental sessions was identical to the one used during familiarization. After the warm-up, the participants started the evaluations. In a randomized, cross-over fashion, the participants performed 3 sets of the BP exercise to volitional failure at a load of 60% 1RM with maximal possible velocity in the concentric and eccentric phases of movement, either preceded by a CA (PAPE) or without activation (CONT). During the CA the participants performed 3 sets of 3 repetitions with a constant duration of 2 s for the eccentric movement and maximal velocity for the concentric phase at a load of 85% 1RM. The rest interval between successive sets equaled 4 min. The time between the CA and post-activation exercise was determined in accordance with the IRT. Every repetition was performed without bouncing the barbell off the chest and without intentionally pausing at the transition between the eccentric and concentric phases. A linear position transducer system (Tendo Power Analyzer, Tendo Sport Machines, Trencin, Slovakia) was used for the evaluation of bar mechanics. The Tendo Power Analyzer is a reliable system for measuring movement velocity and power output [23,24]. The measurement was made independently for each repetition and automatically converted into the values of power output and bar velocity.

The peak power output (PP) and peak bar velocity (PV) were obtained from the best repetition performed in a particular set. The mean of peak power output (PP_MEAN_) and mean of peak bar velocity (PV_MEAN_) were obtained from the peak of all repetitions performed in a particular set. The mean power output (MP) as well as mean bar velocity (MV) were obtained as the mean of all repetitions performed in particular sets. All participants completed the described testing protocol, which was carefully replicated in subsequent experimental sessions. 

### 2.4. Statistical Analysis

Data were presented as the mean ± SD. All variables presented a normal distribution according to the Shapiro-Wilk test. Verification of differences between CONT and PAPE in REP, TUT, PP, PP_MEAN_, MP, PV, PV_MEAN_, and MV was performed using a two-way 2 X 3 (condition x set) analysis of variance (ANOVA) with repeated measures. In cases of a significant main effect, post-hoc comparisons were conducted using Tukey’s test. Percent changes and 95% confidence intervals were also calculated. Effect sizes (Cohen’s *d*) were reported where appropriate and interpreted as large (*d* ≥ 0.80), moderate (*d* between 0.79 and 0.50), small (*d* between 0.49 and 0.20), and trivial (*d* < 0.20) [25]. All statistical analyses were performed using Statistica 9.1 (Hillview Avenue, Palo Alto, CA, USA). The statistical significance was set at *p* < 0.05.

## 3. Results

The two-way repeated measures ANOVA indicated a significant conditions x set main interaction effect for TUT (*p* < 0.01). There was also a significant main effect for the condition in TUT (*p* < 0.01). No statistically significant main effect was revealed for REP, PP, PP_MEAN_, MP, PV, PV_MEAN_, and MV (Table 1). The post-hoc analysis for the main effect indicated significant increases in TUT (*p* < 0.01) for the BP following PAPE, compared to the CONT condition. Finally, there was a significant increase in TUT (*p* < 0.01) in the third set for PAPE compared to the CONT condition (Table 2). The results of the IRT test showed that 4 min was an optimal rest interval for 7 participants, 8 min for 3 participants and 12 min for 2 of them (Figure 1).

## 4. Discussion

The main finding of the study was that the PAPE protocol increased training volume based on TUT in the BP performed to volitional failure compared to CONT conditions. Despite the fact that there was an increase in TUT for the PAPE conditions, the study did not show significant changes in the number of performed REP. Greater TUT, when volume load (REP × load) was equated, may result in greater neuromuscular fatigue, but in our study, we did not observe effects of fatigue reflected by decreasing power output or bar velocity between the PAPE and CONT conditions during the BP exercise performed to volitional failure. Therefore, these results suggest that the PAPE effect can be used to acutely increase the time of effort during resistance exercise, without affecting the concentric bar velocity and power output of each set. 

To the best of the authors knowledge, the presented study is the first to analyze both the changes in training volume as well as in power output and bar velocity following PAPE effects in trained men. Despite the large body of evidence investigating the effect of PAPE on the performance of explosive activities, only two studies have examined the impact of PAPE on resistance training volume [14,15]. However, neither of them analyzed training volume as well as power output and bar velocity simultaneously during sets performed to volitional failure. In terms of the impact of PAPE on training volume based on the number of performed REP, the results obtained in our study contradict those of Sevilmiş and Atalağ [14] and Alves et al. [15], which showed an increase in the number of preformed REP following CA. However, it should be noted that Sevilmiş and Atalağ [14] analyzed only one set of an exercise (compared to three sets in our study), which is not habitual practice during sports training, where several sets of each exercise are used in order to obtain significant adaptive changes. The impact of the PAPE effect on the number of performed REP during more than one set was analyzed only in the study of Alves et al. [15], who showed that the number of performed REP in the first and second sets were greater under PAPE, compared to the control conditions. However, no significant difference was found in the third set. This result is contrary to the presented results, where no significant changes in the number of preformed REP were observed in any of the three sets. Furthermore, the result of effect size showed higher differences between PAPE and CONT conditions in the third set (10.9 ± 2.5 vs. 10.0 ± 2.3 respectively; ES = 0.37) compared to the first (25.8 ± 3.3 vs. 25.8 ± 2.5 respectively; ES = 0) and second set (16.7 ± 2.5 vs. 16.3 ± 2.1 respectively; ES = 0.17), which is opposite to the results obtained by Alves et al. [15]. 

The differences in impact of PAPE on the maximal number of performed REP during sets performed to volitional failure between the presented results and the study of Sevilmiş and Atalağ [14] and Alves et al. [15] may be related to the level of experience in resistance training [8,26], as well as to the movement tempo used during exercise [6,13]. Compared to the presented research, where the participants had a minimum of three years of resistance training experience (6.3 ± 2.1 years), in the study of Sevilmiş and Atalağ [14] and Alves et al. [15], the participants had a minimum of 1 year of resistance training experience. Furthermore, in the study by Sevilmiş and Atalağ [14], as well as that by Alves et al. [15], the movement tempo of each performed REP was controlled and amounted to 2 s in the eccentric phase and 1 s in the concentric phase, while in the present study the participants used maximal tempo of movement. According to Wilk et al. [13], movement tempo has a significant effect on the maximal number of performed REP in a particular set, as well as on the PAPE effect during successive sets of a resistance exercise [6].

Moreover, according to Wilk et al. [13,27] not only the number of performed REP, but also TUT is an important variable to consider when evaluating training volume. However, only one previous study analyzed the impact of PAPE on TUT [14]. Sevilmiş and Atalağ [14] showed a significant increase in TUT for the PAPE condition compared to the control, during the BP exercise performed to volitional failure, which is consistent with the presented study. Furthermore, the presented study showed not only a significant increase of total training TUT, but also a significant increase of TUT in the third set of the BP exercise for the PAPE condition, compared to the CONT. Regardless of the mechanisms underpinning PAPE performance improvement, it can be speculated that the increase in time of the resistance effort could have been obtained as a result of increased muscle temperature and blood flow as well as water content following the CA. Thus, the PAPE protocol may contribute to enhancing the general effects of the warm-up [1]. Furthermore, the increase of TUT during the PAPE condition compared to the CONT (especially in third set) can also be attributed to increased phosphorylation of myosin light chains rendering the actin and myosin molecules more sensitive to Ca^2+^ availability [5], which would allow the participants to maintain a certain amount of force even in the presence of biochemical changes within the working muscle that lead to fatigue. However, an assessment of power output and bar velocity did not show differences between PAPE and CONT conditions, which indicates that the increase in TUT was not related to the changes in concentric bar velocity and power output. Therefore, the increase in TUT with the simultaneous lack of significant differences in the number of performed REP as well as the lack of significant differences in concentric bar velocity and power output indicate that a longer TUT for the PAPE condition was related to the extension of the eccentric phase of movement only.

The present study has several limitations which should be addressed. Although the results showed a significantly greater value of TUT, following the CA, the direct causes of these changes cannot be determined and explained. There was no analysis of direct physiological changes, as well as no electromyography recordings that would be the basis for explaining the obtained results. Since there is no evidence regarding the possible cause for the increased TUT in the PAPE condition, further research is needed to assess the physiological and mechanical variables responsible for these changes.

### Practical Implications

The impact of a CA can been used effectively to extend the duration of effort during resistance exercise performed to volitional failure; therefore, it can be useful in increasing the efficiency of strength-endurance performance. Furthermore, combining several sets of high-load exercises before sets performed to volitional failure can be an effective way to develop both maximal strength and strength-endurance in a single session, without losing the effectiveness of sets whose goal is strength-endurance performance. In addition, such a complex training system can be particularly useful in sports disciplines that, due to a high number of competitions, do not leave adequate time to perform separate training sessions for maximal strength and strength-endurance.

## 5. Conclusions 

In summary, the presented study showed that compared to CONT conditions, PAPE did not significantly affect the number of performed REP during the BP exercise performed to volitional failure. However, an increase in TUT was observed. Even if the applied CA did not increase the training volume (based on REP), it did not adversely affect the level of training volume. Therefore, a training program that combines both a high-load exercise prior to several sets performed to volitional failure could be a valuable alternative, especially for athletes whose sports discipline requires both a high level of muscular power and appropriate strength endurance.

## Figures and Tables

**Figure 1 ijerph-17-02554-f001:**
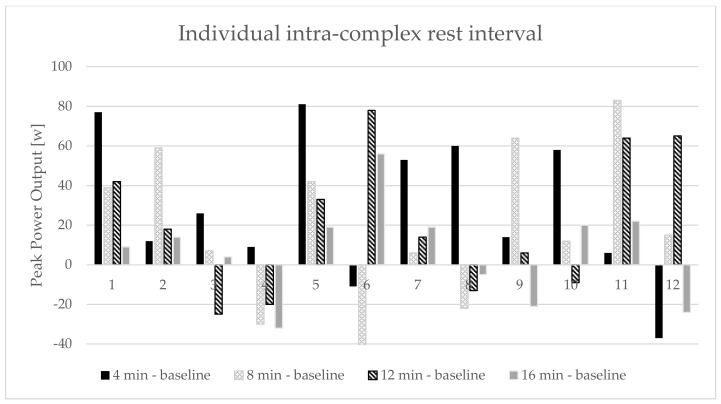
Individual differences in peak power output between baseline and rest interval for each individual during the intra-complex rest interval test. The y-axis represents the difference in peak power output between the baseline and rest interval for each individual.

**Table 1 ijerph-17-02554-t001:** Performance variables during 3 sets of the bench press exercise.

Variables	Conditions (3 Sets)		*p*
	CONTROL	PAPE	
REP [n]	17.4 ± 7.0	17.8 ± 6.8	0.38
TUT [s]	25.1 ± 9.6	26.78 ± 8.3	0.01 *
MP [W]	379 ± 74	385 ± 82	0.13
PP [W]	774 ± 160	727 ± 164	0.18
PP_MEAN_ [W]	570 ± 129	570 ± 127	0.71
MV [m/s]	0.53 ± 0.07	0.52 ± 0.08	0.15
PV [m/s]	0.92 ± 0.14	0.90 ± 0.14	0.14
PV_MEAN_ [m/s]	0.68 ± 0.10	0.68 ± 0.11	0.75

These data present the mean values of the 3 sets. All data are presented as mean ± standard deviation; REP—the number of performed repetitions, TUT—time under tension, MP—mean power output, PP—peak power output, PP_MEAN_—mean of peak power output, MV—mean bar velocity output, PV—peak bar velocity, PV_MEAN_—mean of peak bar velocity; * statistically significant difference *p* < 0.05.

**Table 2 ijerph-17-02554-t002:** Performance variables for particular sets of the bench press exercise.

Performance	Conditions	Set 1 (95% CI)	Set 2 (95% CI)	Set 3 (95% CI)
		REP [n]		
REP [n]	CONTROL	25.8 ± 2.5 (24.3 to 27.4)	16.3 ± 2.1 (14.9 to 17.6)	10.0 ± 2.3 (8.5 to 11.5)
	PAPE	25.8 ± 3.3 (23.6 to 27.9)	16.7 ± 2.5 (15.1 to 18.3)	10.9 ± 2.5 (9.3 to 12.5)
	ES	0	0.17	0.37
TUT [s]	CONTROL	35.3 ± 4.9 (32.2 to 38.3)	25.9 ± 3.4 (23.7 to 28.1)	14.1 ± 3.8 * (11.7 to 16.5)
	PAPE	35.4 ± 5.6 (31.9 to 38.9)	25.8 ± 4.2 (23.1 to 28.4)	19.2 ± 4.8 * (16.1 to 22.2)
	ES	0.02	0.03	1.18
MP [W]	CONTROL	423 ± 68 (380 to 466)	373 ± 68 (330 to 416)	340 ± 68 (297 to 383)
	PAPE	431 ± 71 (386 to 476)	376 ± 75 (328 to 423)	349 ± 84 (296 to 403)
	ES	0.12	0.04	0.12
PP [W]	CONTROL	843 ± 112 (771 to 914)	734 ± 139 (645 to 823)	656 ± 175 (544 to 767)
	PAPE	832 ± 112 (761 to 903)	702 ± 150 (607 to 798)	647 ± 176 (535 to 759)
	ES	0.10	0.22	0.05
PP_MEAN_ [W]	CONTROL	639 ± 99 (576 to 702)	565 ± 115 (492 to 638)	513 ± 145 (421 to 605)
	PAPE	637 ± 108 (569 to 706)	553 ± 128 (472 to 635)	521 ± 125 (441 to 600)
	ES	0.02	0.10	0.06
MV [m/s]	CONTROL	0.59 ± 0.03 (0.57 to 0.62)	0.52 ± 0.05 (0.49 to 0.56)	0.46 ± 0.03 (0.45 to 0.48)
	PAPE	0.60 ± 0.05 (0.56 to 0.63)	0.50 ± 0.04 (0.47 to 0.53)	0.45 ± 0.05 (0.42 to 0.49)
	ES	0.24	0.44	0.24
PV [m/s]	CONTROL	1.05 ± 0.10 (0.99 to 1.11	0.91 ± 0.09 (0.85 to 0.97)	0.81 ± 0.11 (0.74 to 0.88)
	PAPE	1.04 ± 0.09 (0.98 to 1.10)	0.87 ± 0.11 (0.80 to 0.94)	0.80 ± 0.11 (0.73 to 0.87)
	ES	0.11	0.40	0.10
PV_MEAN_ [m/s]	CONTROL	0.79 ± 0.06 (0.75 to 0.83)	0.67 ± 0.08 (0.62 to 0.71)	0.60 ± 0.07 (0.56 to 0.64)
	PAPE	0.78 ± 0.07 (0.74 to 0.83)	0.65 ± 0.08 (0.60 to 0.71)	0.61 ± 0.08 (0.56 to 0.66)
	ES	0.15	0.25	0.13

Data are presented as mean ± standard deviation and 95% confidence interval (95% CI). Effect size (ES) was assessed using Cohen’s *d*. These data present the mean values of the 3 sets. All data are presented as mean ± standard deviation; REP—the number of performed repetitions, TUT—time under tension, MP—mean power output, PP—peak power output, PP_MEAN_—mean of peak power output, MV—mean bar velocity, PV—peak bar velocity, PV_MEAN_—mean of peak bar velocity * statistically significant difference *p* < 0.05.

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
