# Peer review of "Can Post-Activation Performance Enhancement (PAPE) Improve Resistance Training Volume during the Bench Press Exercise?"

_ijerph, 2020, doi:10.3390/ijerph17072554_

Round 1
Reviewer 1 Report
General comments
This is an interesting and original study with the aim of testing the effect of post-activation potentiation on the mechanical responses to a multiple-set BP session. I have minor comments below.
Specific comments
Pg. 2, lines 76-83: Authors should be commended for the original proposal of the study.
Pg. 2, lines 91-92: Why exactly participants were instructed to perform the eccentric phase with maximal contraction velocity? It is not a commonplace in resistance training and can compromise the ecological validity of the study.
Pg. 3, lines 98-100: The participants are strong and probably highly trained. How familiar were them with the experimental training session proposed in the study? In general, strength exercisers at this level do not train with moderate loads, nor struggle to perform at maximal velocities. Hence, it is not easy to them to perform like this.
Pg. 3, lines 128-130: How reliable is this procedure? What is the variation (min and max) of individually determined peak power output?
Pg. 4, table 1: I understand that the difference in TUT is significant. However, do you really consider it meaningful from a practical standpoint?
Pg. 6, lines 226-228: In my opinion, this is the most important finding of the study (which is also meaningful). You could stress more the practical relevance of this finding, not only here, but also in the Practical implications section.
Author Response
General comments
This is an interesting and original study with the aim of testing the effect of post-activation potentiation on the mechanical responses to a multiple-set BP session. I have minor comments below.
Reply - Thank you for your thoughtful and careful review of the manuscript. We believe that your suggestions and our associated revision will significantly improve the quality of the manuscript. You will find your comments below followed by our point-by-point responses in bold.
Specific comments
Pg. 2, lines 76-83: Authors should be commended for the original proposal of the study.
Reply - Thank you for appreciating our work.
Pg. 2, lines 91-92: Why exactly participants were instructed to perform the eccentric phase with maximal contraction velocity? It is not a commonplace in resistance training and can compromise the ecological validity of the study.
Reply - We chose maximum velocity in both phases of the movement, since the aim of this study was to assess the influence of PAP on resistance training volume as well as on power output and bar velocity changes. Previous studies showed that the duration of the eccentric phase affects the level of power output in the concentric part of the lift, thus according to that we decided to use maximum velocities in both phases.
J Sports Sci Med. 2019 Jun 1;18(2):191-197.
Pg. 3, lines 98-100: The participants are strong and probably highly trained. How familiar were they with the experimental training protocol proposed in the study? In general, strength exercisers at this level do not train with moderate loads, nor struggle to perform at maximal velocities. Hence, it is not easy for them to perform like this.
Reply - we agree with the reviewer that the participants well trained strength athletes, however it was a homogeneous group (students or graduates of the Academy of Physical Education, instructors or coaches of resistance training, personal coaches), and they were familiar with different innovative training systems. Thus the fast eccentric and concentric tempo of movement was not to the study participants. Furthermore during the familiarization session, none of the participants reported problems with performing the bench press with maximal movement tempo.
Pg. 3, lines 128-130: How reliable is this procedure? What is the variation (min and max) of individually determined peak power output?
Reply – the procedure was used according to Golas et al. (2016). The results of peak power between baseline and the rest interval was additionally presented new in figure (Line 194).
Pg. 4, table 1: I understand that the difference in TUT is significant. However, do you really consider it meaningful from a practical standpoint?
Reply - from a practical point of view (for most sports) the most important is the time of effort, not the number of performed repetitions, thus from this standpoint the differences are meaningful. However it should be noticed that the analysis of all results (REP, TUT, power, velocity) showed that the increase in TUT occurred to a greater extent by increased TUT in the ECC phase rather than the CON one, what has been explained in the discussion (Line 256 – 262).
Pg. 6, lines 226-228: In my opinion, this is the most important finding of the study (which is also meaningful). You could stress more the practical relevance of this finding, not only here, but also in the Practical implications section.
Reply - According to the suggestion, a new sentence has been added to practical implications. (Line 270 – 272)
Reviewer 2 Report
The authors of the present study have worked diligently on a notable project. Their findings suggest that PAPE increases time under tension, thereby there is an increase in training volume. This supports positive adaptations with this style of training. While I did enjoy reading this manuscript I do have a few suggestions for improvement.
-Line 45. The authors state, ‘Most studies…’ but give only two references. This seems contradictory.
-Line 51. ‘number of repetitions (REP)’. Yet in the rest of the manuscript the authors use, ‘number of performed REPs’ or ‘number of REP’ or ‘number of preformed REP’ (line 205). This particular abbreviation makes reading this manuscript difficult. Do the authors mean REP for just repetition or for the idea of repetitions performed? At times it makes it hard to follow and the language is difficult.
-Line 52. The authors state, ‘TUT is recently being considered’ but one of the references provided is 2006. I wouldn’t call that recent.
-Line 83. One of the strengths of this manuscript is that they used trained individuals. They make this point in the Intro. Accordingly, it would be pertinent to include the population in the hypothesis to make it more specific, and clear to the reader.
-Line 90. Is it ‘subjects’, ‘participants’, or ‘athletes?’ I think I have seen all three. I would choose one and use it consistently for readability.
-Line 100. Where any of the subjects competing? Did all of them know their 1RM, as you have this as an inclusion criteria. What if they failed to lift that in the familiarization, were they excluded?
-Line 120. How many total attempts were allowed? Was there a maximum? Do the authors feel that a one time assessment of the 1RM is efficacious? I think this needs to be addressed in some form or another as you can’t discuss your reliability or validity of your measures with a single assessment.
-Line 159. What software was used for analysis? What was the apriori alpha?
-Line 193. ‘…three studies…’ yet only two references. In addition, only two studies are discussed, so I assume this is a typo.
Author Response
The authors of the present study have worked diligently on a notable project. Their findings suggest that PAPE increases time under tension, thereby there is an increase in training volume. This supports positive adaptations with this style of training. While I did enjoy reading this manuscript I do have a few suggestions for improvement.
Reply - Thank you for your thoughtful and careful review of the manuscript. We believe that your suggestions and our revision will improve the quality of the manuscript. You will find your comments below followed by our point-by-point responses in bold.
-Line 45. The authors state, ‘Most studies…’ but give only two references. This seems contradictory.
Reply - Changes have been made.
-Line 51. ‘number of repetitions (REP)’. Yet in the rest of the manuscript the authors use, ‘number of performed REPs’ or ‘number of REP’ or ‘number of preformed REP’ (line 205). This particular abbreviation makes reading this manuscript difficult. Do the authors mean REP for just repetition or for the idea of repetitions performed? At times it makes it hard to follow and the language is difficult.
Reply - Accordingly, to the Reviewer’s suggestion, we have changed this abbreviation throughout the manuscript.
-Line 52. The authors state, ‘TUT is recently being considered’ but one of the references provided is 2006. I wouldn’t call that recent.
Reply - We agree with Reviewer and decided to remove the statement about the reference “ being recent”.
-Line 83. One of the strengths of this manuscript is that they used trained individuals. They make this point in the Intro. Accordingly, it would be pertinent to include the population in the hypothesis to make it more specific, and clear to the reader.
Reply - changes have been made (Line 88)
-Line 90. Is it ‘subjects’, ‘participants’, or ‘athletes?’ I think I have seen all three. I would choose one and use it consistently for readability.
Reply - According to Reviewer’s suggestion, we have the term participants throughout the whole manuscript.
-Line 100. Where any of the subjects competing? Did all of them know their 1RM, as you have this as an inclusion criteria. What if they failed to lift that in the familiarization, were they excluded?
Reply - It was a homogeneous group (students or graduates of the Academy of Physical Education, instructors or resistance training coaches, personal coaches) which trained on a regular basis before and during the study. All of the participants had their 1RM BP value evaluated within 2-3 weeks of the commencement of the research, and everyone managed to exceed 120% body mass during this lift.
-Line 120. How many total attempts were allowed? Was there a maximum? Do the authors feel that single assessment of the 1RM is efficacious? I think this needs to be addressed in some form or another as you can’t discuss your reliability or validity of your measures with a single assessment.
Reply - The following sentence has been changed (Line 129-130)
-Line 159. What software was used for analysis? What was the apriori alpha?
Reply - The following sentences has been added in the statistical analysis section: “All statistical analyses were performed using Statistica 9.1. The statistical significance was set at p<0.05 (Line 171 – 172)
-Line 193. ‘…three studies…’ yet only two references. In addition, only two studies are discussed, so I assume this is a typo.
Reply - Changes have been made (Line 213)
Reviewer 3 Report
The aim of the study has been clearly defined: how post-activation potentiation enhances resistance training volume in a single session. Overall this is an insightful and novel study that addresses the effects of post-activation phenomenon on training volume. Most previous studies, considering PAP and/or PAPE relate to its effectiveness in generating greater power output or greater force production in explosive type activities, specific for particular sport disciplines. Few research projects have addressed the PAP phenomena on training volume, which is one of the key variables determining the scope of adaptive changes to resistance training. The study provides obvious practical implications for strength and conditioning coaches and practitioners. The authors did a good job addressing the numerous shortcomings of previous research (i.e. measures of kinematic variables, multiple sets after potentiation, individual rest intervals ect.). This applies above all to the simultaneous analysis of variables regarding strength-endurance (REP, TUT), volume of work, as well as power and velocity variables. Besides, new valuable scientific data, the paper includes interesting, and significant practical implications for recreationally active people and competitive athletes.
Title
I suggest a small change in the title, and to add the word “enhancement” : The article could be titled “ Can post-activation potentiation enhancement (PAPE) improve resistance training volume during the bench press exercise?
Abstract:
Line 15: please change to “post-activation potentiation enhancement”
Line 16-17 - please add a space between results and ±
Introduction
Line 45 – velocity of what? Probably movement velocity – please make the appropriate change.
Line 49 – 50 “However, the duration of repetitions preformed in a set, or in a whole training session is not always the same” - please be more specific.
Line 64 – [10] - Incorrect citation – please check throughout the manuscript.
Line 70 – muscular failure – in my opinion it is not muscular failure but volitional failure (as it was described in the first part of the introduction) - please make the appropriate change in the whole manuscript. Line 78 – please add volitional.
Line 80 and 83 – please change muscular to volitional.
Materials and Methods
Line 85 - I do not understand how the crossover design was applied (does the crossover design relate to the familiarization sessions?) - please make the appropriate change.
Line 87 - please add additional IRT information in the introduction section. Line 127 – min. or minute – please unify to minutes.
Line 139 – please add individual before IRT.
Line 148 - please add “output: after power, and that goes for the tables description as well.
Table 2 - I don’t see the sense of writing next to ES - CONT vs PAPE - it's obvious.
Discussion
Line 184 – volume load – you mean reps x load? If yes please add such information in parenthesis.
Line 186: I suggest to write that; power output and bar velocity didn’t change. Since, there were no differences in all power and velocity values. Line 193: The authors mentioned that three studies examined the impact of PAPE on resistance training volume, however you provide only two references.
Line 211 – volitional not muscular.
References – Please check all references for proper abbreviations and citations.
Line 290: IJERPH change to Int J Environ Res Public Health.
Line 292: No authors.
Conclusion: Considering the novel aspects of the manuscript, good methodology, and a well conducted discussion, I recommend to accept the manuscript for print, following the minor corrections suggested in the review.
Author Response
A review of the manuscript submitted to the IJERPH entitled “Can post-activation potentiation (PAPE) improve resistance training volume during the bench press exercise?
The aim of the study has been clearly defined: how post-activation potentiation enhances resistance training volume in a single session. Overall this is an insightful and novel study that addresses the effects of post-activation phenomenon on training volume. Most previous studies, considering PAP and/or PAPE relate to its effectiveness in generating greater power output or greater force production in explosive type activities, specific for particular sport disciplines. Few research projects have addressed the PAP phenomena on training volume, which is one of the key variables determining the scope of adaptive changes to resistance training. The study provides obvious practical implications for strength and conditioning coaches and practitioners. The authors did a good job addressing the numerous shortcomings of previous research (i.e. measures of kinematic variables, multiple sets after potentiation, individual rest intervals ect.). This applies above all to the simultaneous analysis of variables regarding strength-endurance (REP, TUT), volume of work, as well as power and velocity variables. Besides, new valuable scientific data, the paper includes interesting, and significant practical implications for recreationally active people and competitive athletes.
Reply - We would like to thank the Reviewer for the time and effort taken for reviewing our manuscript. We are grateful for the valuable comments and proposals that helped us improve the manuscript. Please find below changes that we have performed in the manuscript according to your suggestions and we hope that our revision meets your expectations.
Title
I suggest a small change in the title, and to add the word “enhancement”: The article could be titled “Can post-activation potentiation enhancement (PAPE) improves resistance training volume during the bench press exercise?
Reply - We agree with Reviewer. Changes have been made
Abstract:
Line 15: please change to “post-activation potentiation enhancement” (Line 14)
Reply - changes have been made.
Line 16-17 - please add a space between results and ±
Reply - Changes have been made.
Introduction
Line 45 – velocity of what? Probably movement velocity – please make the appropriate change.
Reply - Changes have been made (Line 45).
Line 49 – 50 “However, the duration of repetitions preformed in a set, or in a whole training session is not always the same” - please be more specific.
Reply - We agree with Reviewer. The sentence has been corrected (Line 49-52)
Line 64 – [10] - Incorrect citation – please check throughout the manuscript.
Reply - References has been corrected (Line 64)
Line 70 – muscular failure – in my opinion it is not muscular failure but volitional failure (as it was described in the first part of the introduction) - please make the appropriate change in the whole manuscript.
Reply - Changes have been made throughout the manuscript.
Line 78 – please add volitional.
Reply - Changes have been made.
Line 80 and 83 – please change muscular to volitional.
Reply - Changes have been made.
Materials and Methods
Line 85 - I do not understand how the crossover design was applied (does the crossover design relate to the familiarization sessions?) - please make the appropriate change.
Reply - The crossover design was related only to the experimental sessions. The sentence has been changed (Line 93 – 97)
Line 87 - please add additional IRT information in the introduction section.
Reply - According to Reviewer’s suggestion, the following sentences have been added to the introduction (Line 75 – 81)
Line 127 – min. or minute – please unify to minutes.
Reply - Appropriate changes were made.
Line 139 – please add individual before IRT.
Reply - Changes have been made.
Line 148 - please add “output: after power, and that goes for the tables description as well.
Reply - Changes have been made.
Table 2 - I don’t see the sense of writing next to ES - CONT vs PAPE - it's obvious.
Reply - We agree with Reviewer, “CONT vs PAPE” has been removed.
Discussion
Line 184 – volume load – you mean reps x load? If yes please add such information in parenthesis.
Reply - Changes have been made, we added “(repetitions x load)” (Line 204)
Line 186: I suggest to write that; power output and bar velocity didn’t change. Since, there were no differences in all power and velocity values.
Reply - Changes have been made
Line 193: The authors mentioned that three studies examined the impact of PAPE on resistance training volume, however you provide only two references.
Reply - The sentence has been corrected (Line 213)
Line 211 – volitional not muscular.
Reply - Changes have been made
References – Please check all references for proper abbreviations and citations.
Line 290: IJERPH was changed to Int J Environ Res Public Health.
Line 292: No authors.
Reply - Changes have been made and whole references section corrected.